# Multi-Objective Optimization of a Microgrid Considering the Uncertainty of Supply and Demand

Shiping Geng [1,2,*], Gengqi Wu [1,2], Caixia Tan [1], Dongxiao Niu [1,2] and Xiaopeng Guo [1]

1   School of Economics and Management, North China Electric Power University, Beijing 102206, China; wugengqi@ncepu.edu.cn (G.W.); 120192206106@ncepu.edu.cn (C.T.); niudx@126.com (D.N.); 13520328997@163.com (X.G.)
2   Beijing Key Laboratory of New Energy and Low-Carbon Development, Beijing 102206, China
*   Correspondence: Gengsp@ncepu.edu.cn; Tel.: +86-152-1015-1580

**Abstract:** Starting from the perspective of the uncertainty of supply and demand, using the Copula function and fuzzy numbers a scenario generation method, considering the uncertainty of scenery, and a random fuzzy model of energy demand uncertainty are proposed. Then, through the energy flow direction and the energy supply, production, conversion, storage, and demand, a multi-objective model considering the economic and environmental protection of a park is constructed. Here, the park refers to a microgrid that gathers distributed energy such as wind and photovoltaics and has requirements for cooling, heat, and electricity at the same time. Next, combining the constraints of each link, the particle swarm algorithm is used to solve the model. Finally, an example is analyzed in a certain park. The results of the example show that, on the one hand, the proposed scenario generation method and fuzzy number method can reduce the uncertainty of supply and demand, effectively fitting the wind and photovoltaic output and various energy demands. On the other hand, considering the economy and environmental protection of the park at the same time, the configuration of energy storage equipment can not only improve the economy of the park, but also promote the consumption of renewable energy.

**Keywords:** complementarity; supply and demand uncertainty; copula function; multi-objective optimization

## 1. Introduction

In recent years, under the guidance of policy, the wind and solar renewable energy sector in China has been vigorously developed to solve the energy problem, but the large-scale development of renewable energy has caused a large number of wind and light abandonment problems [1]. The microgrid integrates distributed energy, which can realize the complementary utilization of renewable energy such as wind and solar energy and become an important means to promote the consumption of renewable energy [2]. However, due to the uncertainty of renewable energy output and the subjectivity of user energy consumption, the operation of a multienergy complementary system is limited. Therefore, the park considers uncertain multienergy complementary scheduling optimization, which has become a current research focus.

Regarding the processing of multienergy complementary uncertainty, robust optimization and stochastic programming are mainly used to deal with load uncertainty at home and abroad [3]. Yang et al. [4] used the box uncertainty set in stochastic programming to describe load uncertainty. Shen et al. [5] and Parisio A et al. [6] used robust optimization to reduce load uncertainty. Shen et al. [5] used the k-means clustering method to generate typical daily load scenarios and used the upper and lower ranges to describe the load end uncertainty to build a robust optimization model. Due to the conservative decision-making of the robust planning method, Gong et al. [7] used the Fourier fitting

method to fit various loads at the end of the load and verify that the fitting method was applicable. For supply uncertainty, the scenario generation method and robust optimization have been introduced to reduce the uncertainty of wind and photovoltaic output. Kong et al. [8] and Wang et al. [9] proposed a robust stochastic optimization scheduling method for multiple uncertain scheduling problems in multienergy virtual power plants. Li et al. [10], taking into account with the uncertainty of distributed energy output in a multienergy complementary park and the differences in the flow of cooling, heat, and electric energy, proposed an improved universal generating function (UGF) method to improve the reliability of the energy supply system in the park. Ana Turk et al. [11] proposed a two-stage stochastic dispatching scheme for a comprehensive multienergy system, using corresponding probabilities to generate specific actual scenarios to represent the randomness of wind power uncertainty and, ultimately, greatly improve the economic benefits of the system and the utilization of the wind power rate. Li et al. [12] used the scenario reduction method to construct a day-ahead dispatch model considering wind power fluctuations. It can be seen that the existing studies have more characterizations of the unilateral uncertainty of the supply side or the demand side, and there are fewer studies considering the uncertainties of both the supply and demand sides or multiple supplies and multiple demands.

More and more scholars at home and abroad have begun to study effective methods to deal with the uncertainty of the multienergy complementary park system and optimize the problem [13,14]. Regarding the handling of the uncertainty of multienergy complementarity, most studies use robust optimization for the final solution of the model [15,16], but this method has certain limitations in showing the correlation between two or more variables. Facing the random correlation between wind speed, sunshine, and load power, the Copula function can more accurately describe the correlation structure of multivariate variables, and is widely used in the modeling of the dependent structure of two (or more) random variables. Wang et al. [17] and Amir Aris Lekvan et al. [18] both proposed that the power demand and transportation mode of plug-in electric vehicles based on the Copula function model can be embedded in the planning of the probability distribution system. Valizadeh Haghi H et al. [19] investigated an integration study of photovoltaics and wind turbines distributed in a distribution network based on stochastic modeling using Archimedean copulas as a new efficient tool. In view of the uncertainty of energy consumption on the user side, traditional uncertain processing methods, such as the stochastic programming model [4], cloud model theory [20], and interval estimation [21], ignore the economic impact on the process of processing—that is, users participate in the demand response under different price signals or compensation policies. The uncertainty of the response load is directly related to the compensation level. Using the random fuzzy function to describe user load can effectively avoid this problem and reduce the deviation between the predicted value and the actual value. Cui et al. [22] introduced the fuzzy chance constraint, relaxed the deterministic constraint into the system constraint with fuzzy variables, and made it clear by using trapezoidal fuzzy parameters. In order to reflect these uncertainties, Ji et al. [23] proposed a hybrid inexact stochastic fuzzy chance constrained programming (ITSFCCP).

In research on multienergy complementary scheduling, the objective function is to optimize the scheduling with the minimum operating cost or maximum profit of the system. Feng et al. [24] and Ju et al. [25] constructed a day-ahead and real-time optimal scheduling model with the least operating cost in the decision-making stage. Chen et al. [26] considered the integrated demand response and the uncertainty of wind output and discussed the optimization of the coupled heat–power–gas (CHPG) microgrid with the goal of minimizing the operating cost and risk of the wind power grid-connected microgrid. Aiming at the intermittency and instability of wind and solar energy and the easy compensation of hydropower stations, Liu et al. [27] proposed a wind-solar hydropower optimal dispatch model with the goal of maximizing total system power generation and minimizing the 10-day combined power generation. However, with policy

guidance, the multienergy complementary system is no longer suitable for pursuing a single economic goal, and it has begun to pursue multi-objective optimization decisions. Zhu et al. [28] established a multi-objective optimization model that simultaneously optimizes the economic benefits and operational safety of the hybrid power system. Ju et al. [29] proposed a virtual power plant (VPP) considering an operation flexible risk avoidance model with the goal of maximizing operating profit and minimizing operating risk in order to realize the optimal operation of a virtual power plant. Considering the operation efficiency of generating units, the dissatisfaction of the demand response, and total profit as the objectives, Wang et al. [30] studied the optimization of load management in a micro energy grid. However, there are few documents that unify the relatively opposed goals of economy and environmental protection at the same time so as to optimize system scheduling. With the "Carbon Peak" and "Carbon Neutral" goals proposed, the issue of greenhouse gas emissions in the power generation process has attracted much attention. Therefore, it is necessary to study the environmental issues of the multienergy complementary park.

Therefore, based on the above research, this article first uses the Frank function in the Copula function to construct a wind-solar complementary joint-distribution function and then uses the second-order Fourier fitting method to fit the energy demand, such as cooling, heating, and electricity, to generate a scenario of typical daily wind and solar output and energy demand to solve the problem of considering only one-sided uncertainty. Finally, in order to consider the economy and environmental protection of the system at the same time, a multiobjective multienergy complementary park scheduling optimization model was proposed with the largest benefit of the multienergy complementary park and the lowest rate of outsourced electricity to promote the consumption of renewable energy. The contributions of the study are as follows:

(1) The Copula function was used to generate a wind and solar complementary joint-distribution function to simulate the uncertainty of wind power and photovoltaic output, and the typical daily scenario generated can more effectively fit the wind power and photovoltaic output characteristics of the park.

(2) A random fuzzy model is constructed based on fuzzy numbers to fit the park's cooling, heating, and power loads; describe user loads; and reduce the deviation between the predicted value and the actual value.

(3) From the economic and environmental protection aspects of the multienergy complementary park, this paper constructs the objective function pursued by the park to improve the park's income, promote the consumption of renewable energy, and realize multienergy complementary optimal scheduling within the park.

## 2. Microgrid

### 2.1. A Scenario Generation Based on the Uncertainty of the Scenario

Due to the natural attributes of wind power and solar energy, wind power and photovoltaic power generation have volatility and randomness. In order to ensure the safe and stable operation of the park system, a scenario generation method considering the uncertainty and correlation of wind and solar energy is proposed based on the Copula theory, and typical daily wind and solar output curves with time series are obtained [31,32]. The specific generation steps are as follows:

(1) Calculation of the kernel density estimation function.

The construction of the Copula function depends on correlation coefficients and marginal distribution functions. The methods of selection of correlation coefficients include two-stage estimation, maximum likelihood estimation based on the empirical distribution, and maximum likelihood estimation based on the non-parametric kernel density. Since the total output sequence of wind and photovoltaic power plants is huge and the location parameters in the marginal distribution function cannot be accurately obtained, the non-parametric kernel density maximum likelihood estimation method using

a semiparameter, which does not need to obtain the marginal distribution parameter, can improve the accuracy of the Copula function construction. Based on the historical output sequence of wind power and photovoltaic energy, the output sequences of wind power and photovoltaic power are, respectively, $m = (m_1, m_2, \cdots m_t, \cdots, m_{24})$ and $n = (n_1, n_2, \cdots n_t, \cdots, n_{24})$. Accordingly, the estimation functions of wind power and photovoltaic kernel density are as follows:

$$f(x_1) = \frac{1}{T} \sum_{t=1}^{T} R_e(x_1 - m) \tag{1}$$

$$f(x_2) = \frac{1}{T} \sum_{t=1}^{T} R_e(x_2 - n) \tag{2}$$

where $f(x_1)$ and $f(x_2)$ represent the marginal distribution functions of wind power and photovoltaic power, respectively. T is the capacity sequence and $R_e$ is the kernel function.

(2) Calculation of the correlation coefficient.

The maximum likelihood estimation function and the correlation coefficient of wind power and photovoltaic are as follows:

$$L(\rho) = \sum Inc(f(x_1), f(x_2)) \tag{3}$$

$$\hat{\rho} = argmaxL(\rho) \tag{4}$$

where $\hat{\rho}$ is the estimation of the correlation coefficient.

(3) Generation of the wind–solar complementary joint distribution function.

Copula functions include normal distribution, t-distribution, Gumbel, Clayton, and Frank. There is a negative correlation between wind power generation and photovoltaic power generation. Gumbel and Clayton cannot describe the negative correlation, so the Frank function was selected. Substituting the above historical output series and the correlation coefficient into the Frank function, the wind–solar complementary joint distribution function is as follows:

$$F_F(m, n/\rho) = -\frac{1}{\rho} In(1 + \frac{(e^{-m} - 1)(e^{-n} - 1)}{e^{-\rho} - 1}) \tag{5}$$

(4) Generation of typical output scenarios.

Firstly, according to the wind and solar complementary joint distribution function of each period, random sampling is carried out. Secondly, based on the random sampling and the distribution function, the wind and solar power output of each period were obtained. As a result, the typical daily wind and solar output sequence and curve are, respectively, $x_1 = (x_{1\_1}, x_{1\_2}, \cdots, x_{1\_t}, \cdots, x_{1\_24})$ and $x_2 = (x_{2\_1}, x_{2\_2}, \cdots, x_{2\_t}, \cdots, x_{2\_24})$.

### 2.2. Stochastic Fuzzy Model Based on Load Uncertainty

The cooling heating and power loads of the park are affected by the subjectivity of users, which has uncertainty and fuzziness. Therefore, the random fuzzy function is used to describe the user load to reduce the deviation between the predicted result and the actual result [33,34]. First of all, in order to simplify the calculation and universality of the model, second-order Fourier is used to fit all kinds of load distribution.

$$\begin{cases} f_j(y) = c_0 + \sum_{i=1}^{2} [c_{ij} cos(ijwy) + d_{ij} sin(ijwy)] \\ f_{j\_min}^t \leq y \leq f_{j\_max}^t (j = 1,2,3) \end{cases} \tag{6}$$

where $f^t_{j\_min}$ and $f^t_{j\_max}$ are the minimum and maximum values in the historical data of the class $j$ load collected and $j = 1$ is the electrical load, $j = 2$ is the cooling load, and $j = 3$ is the heating load.

Then, the membership degree of the second-order Fourier parameter series is fitted with the improved first-order Gaussian function:

$$u(y) = \begin{cases} 0 & y < y_{min}, y > y_{max} \\ e^{-(\frac{y-\lambda_1}{\lambda_2})^2} & y_{min} \leq y \leq y_{max} \\ 1 & y = y_h \end{cases} \tag{7}$$

In Equation (7), $\lambda_1$ and $\lambda_2$ are Gaussian function parameters; $y_{min}$ and $y_{max}$ are the minimum and maximum values of the above parameter columns; and $y_h$ is the parameter value when the Gaussian function takes the peak value. After the parameters were fuzzified by the Gaussian function, the fuzzy second-order Fourier function was obtained as follows:

$$f_j(\zeta_y) = \zeta_{c_0} + \sum_{i=1}^{2}[\zeta_{c_{ij}}cos(ij\zeta_w\zeta_y) + \zeta_{ij}sin(ij\zeta_w\zeta_y)] \tag{8}$$

where $\zeta_{c_0}$, $\zeta_{c_{ij}}$, and $\zeta_{ij}$ are the fuzzy variables of the parameter series.

## 2.3. Analysis of Microgrid

The multienergy complementary park system includes five links: energy supply, production, conversion, storage, and demand. The energy supply includes wind energy, solar energy, natural gas, and other energy inputs. Wind energy and solar energy are input into wind turbines and photovoltaic units to generate electricity through the production process. Natural gas is imported into gas turbines and gas-fired boilers to generate electric energy and heat energy. When the electric energy is insufficient, electricity is purchased from the higher-level power grid. In the conversion process, all kinds of energy can be converted, including electricity to gas, electricity to heat, electricity to cooling, and heat to cooling. Energy storage includes heat storage, cooling storage, gas storage, and electricity storage. When the supply of various loads exceeds the demand, the energy is stored and released when the supply is less than the demand. The framework of the multienergy complementary park is shown in Figure 1.

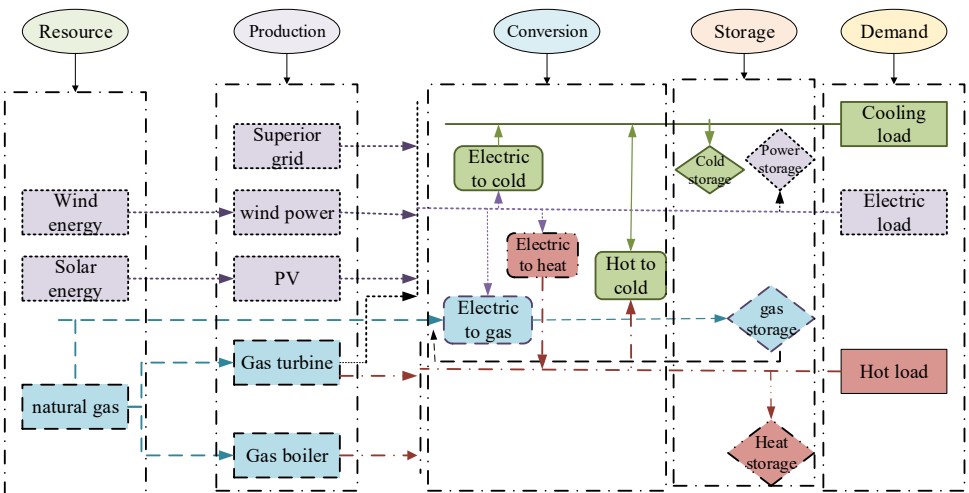

**Figure 1.** Framework of the multienergy complementary park.

According to the specific flow of energy in the multienergy complementary frame diagram, the energy model of multienergy complementary park system can be constructed.

(1) Modeling of energy production.

Wind power and photovoltaic power can be generated according to Equations (1)–(5). Gas turbines consume natural gas to provide power and heat energy, and their energy modeling is as follows:

$$
\begin{cases}
P_{CGT,t} = Q_{CGT,t}\lambda_{ng}\beta_{CGT,p} \\
H_{CGT,t} = Q_{CGT,t}(1 - \beta_{CGT,P} - \beta_{CGT,loss})\beta_{CGT,H}
\end{cases}
\tag{9}
$$

where $P_{CGT,t}$ and $H_{CGT,t}$ are the electric power and thermal power produced by the gas turbine at time $t$. $Q_{CGT,t}$ is the natural gas consumed by the gas turbine at time $t$. $\lambda_{ng}$ is the calorific value of natural gas. $\beta_{CGT,p}$, $\beta_{CGT,loss}$, and $\beta_{CGT,H}$ are the power generation efficiency, the energy consumption efficiency, and the heating efficiency of the gas turbine. The gas-fired boiler in energy production is mainly used for heating, and the heating modeling is as follows:

$$
H_{GB,t} = Q_{GB,t}\lambda_{ng}\beta_{GB},
\tag{10}
$$

where $H_{GB,t}$ is the heating power of gas-fired boilers and $\beta_{GB}$ is the heat production efficiency of gas-fired boilers.

(2) Modeling of energy conversion.

The energy conversion link is that the electricity and heat generated by wind turbines, photovoltaic generators, gas turbines, and gas-fired boilers are converted into other forms of energy through converters. The conversion modeling is as follows:

$$
\begin{bmatrix}
G_{P2G,t} \\
C_{P2C,t} \\
H_{P2H,t} \\
C_{H2C,t}
\end{bmatrix}
=
\begin{bmatrix}
P_{P2G,t} & 0 & 0 & 0 \\
0 & P_{P2C,t} & 0 & 0 \\
0 & 0 & P_{P2H,t} & 0 \\
0 & 0 & 0 & H_{H2C,t}
\end{bmatrix}
\begin{bmatrix}
\gamma_{P2G} \\
\gamma_{P2C} \\
\gamma_{P2H} \\
\gamma_{H2C}
\end{bmatrix}
\tag{11}
$$

In Equation (11), $G_{P2G,t}$, $C_{P2C,t}$, $H_{P2H,t}$, and $C_{H2C,t}$ are, respectively, the natural gas of electricity to gas, the cooling power of electric cooling, the thermal power of electricity to heat, and the cooling power of hot to cooling at time $t$. $P_{P2G,t}$, $P_{P2C,t}$, and $P_{P2H,t}$ are the electric power consumed by electricity to gas, electricity to cooling, and electricity to heat. $H_{H2C,t}$ is the hot consumed by hot to cooling conversion.

(3) Modeling of energy storage.

Energy storage includes energy storage and release. The electric energy storage is charged at the low load and discharged at the peak load by mechanical, electromagnetic, and chemical methods. Thermal energy storage equipment is used for heat storage and release, and includes thermal storage tanks and thermal storage beds. The operation mechanism of the above energy storage modes is the same, and the operation model is as follows:

$$
\begin{cases}
P_{ES,t} = \omega_{ES,t}^{ch}P_{ES,t}^{ch} - \omega_{ES,t}^{dis}P_{ES,t}^{dis} \\
S_{ES,t} = (1 - \beta_{ES,t}^{loss})S_{ES,t-1} + (P_{ES,t}^{ch}\gamma_{ES,t}^{ch} - P_{ES,t}^{dis}/\gamma_{ES,t}^{dis})
\end{cases}
\tag{12}
$$

where $P_{ES,t}$ is the net output power of the energy storage equipment at time $t$, $\omega_{ES,t}^{ch}$ and $\omega_{ES,t}^{dis}$ are Boolean variables, $P_{ES,t}^{ch}$ and $P_{ES,t}^{dis}$ are the energy storage and the release of energy storage equipment at time $t$, $S_{ES,t}$ is the energy storage of the energy storage

equipment at time $t$, and $\gamma_{ES,t}^{ch}$ and $\gamma_{ES,t}^{dis}$ are the energy storage and release efficiency of the energy storage equipment at time $t$.

(4) Modeling of energy demand.

Through energy production, conversion, and storage, it can supply power, cooling, and heating energy demand of internal users in the park. According to the energy flow balance, the energy modeling of the user demand link is as follows:

$$\begin{cases} P_{pv,t} + P_{w,t} + P_{grid,t} + P_{CGT,t} - P_{P2G,t} - P_{P2C,t} - P_{P2H,t} + \Delta P_{CGT,t} = P_{load} \\ H_{CGT,t} + H_{GB,t} - H_{H2C,t} + H_{P2H,t} + H_{ES,t} + \Delta H_{CGT,t} = H_{load} \\ C_{P2C,t} + C_{H2C,t} + C_{ES,t} = C_{load} \end{cases} \quad (13)$$

## 3. Multi-Objective Optimization of Microgrid

### 3.1. An Optimization Model

Based on the above scenario of the generation of wind power and photovoltaic uncertainty and random fuzzy functions, the scheduling optimization strategy for the microgrid is formulated. When the actual output of the park is less than the actual energy demand, the energy storage system will first release energy to meet various kinds of energy demand shortages. If the energy storage system cannot meet the energy demand shortage, it can be purchased from the external network. When the actual output of the park is greater than the energy demand, the energy storage equipment will store energy. The park scheduling strategy is shown in Figure 2.

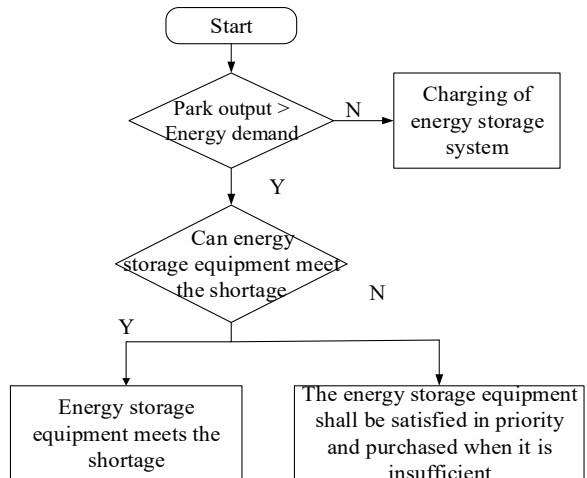

**Figure 2.** Multienergy complementary park scheduling strategy diagram.

(1) Objective functions of scheduling optimization.

This article assumes that the purchased electricity is supplied by conventional generators such as thermal power generators. Therefore, under the condition of certain energy demand, the less electricity is purchased, the cleaner the energy that the system consumes is. The goal of a microgrid is to maximize revenue and minimize the amount of electricity purchased to promote the consumption of clean energy and aid sustainability.

$$\begin{cases} g_1 = maxR_{total} = \sum_{t=1}^{T}(R_{EP,t} + R_{EC,t} + R_{ES,t}) \\ R_{EP,t} = R_{wpp,t} + R_{pv,t} + R_{CGT,t} + R_{GB,t} \\ R_{CGT,t} = P_t^e P_t + P_t^h H_{CGT,t} - P_{ng,t}Q_{CGT,t} \\ R_{GB,t} = P_t^h H_{GB,t} - P_{ng,t}Q_{GB,t} \\ R_{EC,t} = P_{EC,t}^{out}E_{EC,t}^{out}\gamma_{EC,t}^{out} - P_{EC,t}^{in}E_{EC,t}^{in} / \gamma_{EC,t}^{in} \\ R_{ES,t} = P_{ES,t}^{out}E_{ES,t}^{out}\gamma_{EC,t}^{out} - P_{ES,t}^{in}E_{ES,t}^{in} / \gamma_{EC,t}^{in} \end{cases} \quad (14)$$

In Equation (14), $R_{total}$ is the total income of the microgrid. $R_{EP,t}$ is the income of the production link, including wind turbines, photovoltaic units, gas turbines, and gas boilers. $P_t^e$, $P_t^h$, and $P_{ng,t}$ are the prices of power supply, heating, and natural gas at time $t$. $R_{EC,t}$ is the income from the conversion link, including the income of electricity to gas, electricity to heat, electricity to cooling, and heat to cooling. $P_{ES,t}^{out}$ and $E_{ES,t}^{out}$ are the energy supply price and the energy supply of storage equipment at time $t$. $P_{ES,t}^{in}$ and $E_{ES,t}^{in}$ are the energy price and energy consumption of storage equipment at time $t$. The objective function of minimum electricity purchase in the park is as follows:

$$g_2 = \min \theta = \frac{Q_{grid}}{Q_{EP} + Q_{EC} + Q_{ES}}, \quad (15)$$

where $\theta$ is the power purchase rate of the park, $Q_{grid}$ is the electricity purchased from the grid, $Q_{EP}$ is the power generation of the production link, $Q_{EC}$ is the electricity quantity converted in the conversion link, and $Q_{ES}$ is the discharge electricity of the storage link.

Due to the conflict between the maximum profit of the park system and the minimum purchased electricity, the comprehensive weight method of the entropy weight method and the sequence relation analysis method [35] was used to set the weight, which can reduce the influence of subjective factors, reduce the weight setting error, and convert the multi-objective function into a single objective function.

$$g = \lambda_1 g_1 + \lambda_2 g_2 \quad (16)$$

(2) Constraints of scheduling optimization.

The constraints of the production link include the upper and lower limits of the output:

$$Q_{EP}^{min} \leq Q_{EP} \leq Q_{EP}^{max}, \quad (17)$$

where $Q_{EP}^{min}$ and $Q_{EP}^{min}$ are the minimum and maximum values of the output.

The constraints of the conversion link include the upper and lower limits of energy consumption and energy supply:

$$\begin{cases} \sigma_{EC,t}^{out}E_{EC,t}^{out,min} \leq E_{EC,t}^{out} \leq \sigma_{EC,t}^{out}E_{EC,t}^{out,max} \\ \sigma_{EC,t}^{in}E_{EC,t}^{in,min} \leq E_{EC,t}^{in} \leq \sigma_{EC,t}^{in}E_{EC,t}^{in,max} \\ \sigma_{EC,t}^{in} + \sigma_{EC,t}^{out} \leq 1 \end{cases}, \quad (18)$$

where $\sigma_{EC,t}^{out}$ and $\sigma_{EC,t}^{in}$ are the state variables of energy release and storage. The third formula in Equation (18) ensures that the energy storage and release are carried out simultaneously. The operation constraints of the storage link include the upper and lower limits of energy storage and released, the upper and lower limits of power, and the upper and lower limits of capacity.

$$\begin{cases} \sigma_{ES,t}^{ch} Q_{ES,t}^{ch,min} \leq Q_{ES,t}^{ch} \leq \sigma_{ES,t}^{ch} Q_{ES,t}^{ch,max} \\ \sigma_{ES,t}^{dis} Q_{ES,t}^{dis,min} \leq Q_{ES,t}^{dis} \leq \sigma_{ES,t}^{dis} Q_{ES,t}^{dis,max} \\ \sigma_{ES,t}^{ch} + \sigma_{ES,t}^{dis} \leq 1 \\ S_{ES,t}^{min} \leq S_{ES,t} \leq S_{ES,t}^{max} \\ S_{ES,T_0} = S_{ES,T} \end{cases} \quad (19)$$

In Equation (19), $\sigma_{ES,t}^{ch}$ and $\sigma_{ES,t}^{dis}$ are the start and stop states of energy storage and release at time $t$. $Q_{ES,t}^{ch,min}$ and $Q_{ES,t}^{ch,max}$ are the minimum and maximum power of energy storage. $Q_{ES,t}^{dis,min}$ and $Q_{ES,t}^{dis,max}$ are the minimum and maximum power of energy release. $S_{ES,t}^{min}$ and $S_{ES,t}^{max}$ are the minimum and maximum values of energy storage at time $t$.

### 3.2. The Algorithm

The solution model is a non-linear programming model. Particle swarm optimization has the ability to perform autonomous decision-making and distributed decision-making to seek the global optimal solution. Therefore, this paper used particle swarm optimization to solve the problem [36]. The specific solution flow is shown in Figure 3.

Based on the particle swarm optimization algorithm, the specific solving steps of the model are as follows:

Step 1: $u$ typical daily wind and solar output scenarios are generated according to Equations (1)–(5), and then the fuzzy process is carried out for various energy demands according to Equations (6)–(8).

Step 2: The population that contains $n$ chromosomes is initialized.

Step 3: According to the optimal scheduling strategy of different scenarios, whether the maximum number of iterations is reached is determined. If the maximum number of iterations is reached, the optimal scheduling strategy is output; otherwise, the algorithm proceeds to step 4.

Step 4: Based on the scenario strategy optimization, the initial optimization strategy is generated, including the initial energy storage and the purchased electricity.

Step 5: Particle swarm optimization is used to analyze the influence of strategy selection on the comprehensive objective function.

Step 6: When the park strategy changes, based on the cross analysis the energy storage and purchased electricity should be adjusted to ensure that the optimal current strategy is the best.

Step 7: A new generation population is generated, and then steps 3–6 are repeated until the maximum number of iterations is reached and the optimal strategy is output.

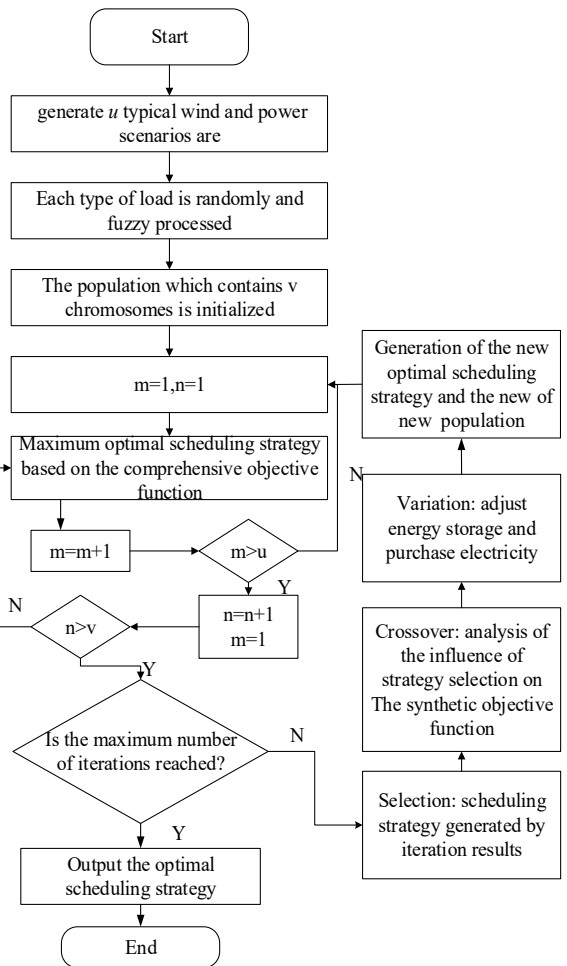

**Figure 3.** Model specific solution process**.**

## 4. Example Analysis

### 4.1. Basic Data

Taking a park as an example, the effectiveness of the proposed model was verified. The park is equipped with wind turbines, photovoltaic units, gas turbines, and a gas boiler, with capacities of 1000 kW, 500 kW, 2500 kW, and 2500 kW, respectively. The conversion link was equipped with the equipment of electricity to cooling, electricity to hot, electricity to gas, and hot to cooling, with capacities of 1000 kW, 1000 kW, 1000 kW, and 1000 kW, respectively. The storage link was equipped with ice storage tanks for cooling storage, thermal energy storage equipment for heat storage, batteries for power storage, and gas storage tanks for gas storage, with capacities of 1500 kWh, 1500 kWh, 1500 kWh, and 500 m³, respectively. It is assumed that the operating efficiency, conversion, and storage of energy production was 95%. Referring to reference [37], the specific operating parameters of the equipment are set as shown in Table 1.

**Table 1.** Equipment operating parameters.

| Equipment | Parameter | Parameter value | Parameter | Parameter Value |
|---|---|---|---|---|
| Ice storage tank | Maximum value of energy storage | 750 kW | Maximum value of energy release | 750 kW |
| Thermal energy storage | Maximum value of energy storage | 750 kW | Maximum value of energy release | 750 kW |
| Battery | Maximum value of electricity storage | 750 kW | Maximum value of electricity release | 750 kW |
| Gas storage tanks | Maximum value of gas storage | 100 m³ | Maximum value of gas release | 100 m³ |
| Gas to electricity equipment | Maximum value of electricity to gas | 1000 kW | Maximum value of gas to electricity | 1000 kW |
| Electricity to heat equipment | Maximum value of electricity to hot | 1000 kW | Maximum value of hot to electricity | 1000 kW |
| Electricity to cooling equipment | Maximum value of electricity to cooling | 1000 kW | Maximum value of cooling to electricity | 1000 kW |
| Hot to cooling equipment | Maximum value of hot to cooling | 1000 kW | Maximum value of cooling to hot | 1000 kW |
| Natural gas | Calorific value of natural gas | 10.01 kWh/m³ | | |

The wind speed and light intensity of the park in 2019 are shown in Figure 4. Among them, the X-axis label of Figure 4a is the number of hours in a year, a total of 8760 h, and the Y-axis label is the wind speed at different times. The X-axis label of Figure 4b is also the number of hours in a year, a total of 8760 h, and the Y-axis is the light intensity at different times.

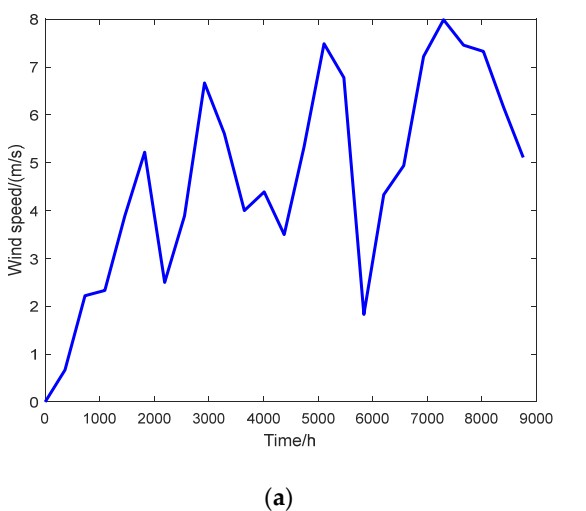

(**a**)

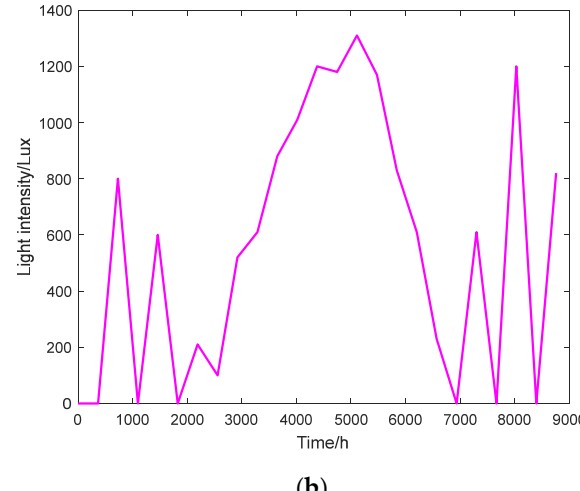

(**b**)

**Figure 4.** (**a**): Wind speed in 2019 and (**b**): light intensity in 2019.

In reality, the price of natural gas is relatively fixed, but in order to fully reflect the dispatching operation and energy flow in the system, similar to the price of electricity, heat, and cooling, this article considered that natural gas also has a time-of-use price. Combined with reference [38,39], the energy selling prices of electricity, heat, cooling, and gas in different periods of time in the park are shown in Figure 5. At the same time, particle swarm optimization was used to solve the model. The initial population size was 200 and the maximum iteration number was 500.

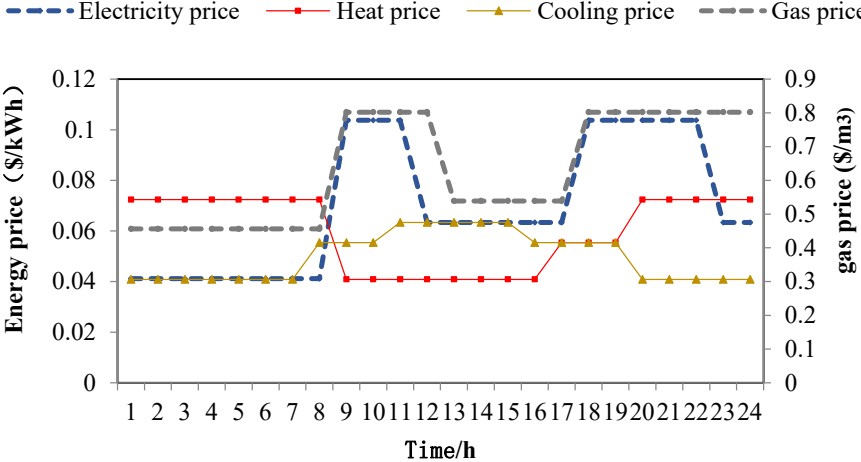

**Figure 5.** Different load selling prices.

*4.2. Result Analysis*

(1) Complementary results analysis.

　　Considering the goal of maximizing profit and minimizing the purchase of electricity, we analyzed the correlation between wind power and photovoltaic power generation firstly. Then, we further analyzed the correlation between wind power generation, photovoltaic power generation, and total energy demand. Finally, we analyzed the correlation between electricity demand and electricity price.

　　(1) Analysis of correlation results between wind power generation and photovoltaic power generation.

　　In order to verify the effectiveness of the joint complementary distribution function model selected in this article, Spearman's correlation coefficient, Kendall's correlation coefficient, Euclidean distance, and maximum distance were used as four evaluation indicators. Among them, Spearman's correlation coefficient measures the degree of linear correlation of variables and Kendall's correlation coefficient measures whether the variable changes are consistent. The closer these two indicators are to the variable empirical data, the better the fitting effect of the modeling will be. The Euclidean distance and the maximum distance measure the degree of difference between the model built and the variable empirical data. The smaller the value, the smaller the difference and the higher the model fit. We calculated the above four index values of the five Copula functions of normal, t-distribution, Gumbel, Clayton, and Frank as shown in Table 2.

**Table 2.** Five kinds of Copula function evaluation index values.

| | Spearman Correlation Coefficient | Kendall Correlation Coefficient | Euclidean Distance | Maximum Distance |
|---|---|---|---|---|
| sample | −0.2903 | −0.2114 | — | — |
| normal | −0.2603 | −0.1569 | 0.6205 | 0.0683 |
| t-distribution | −0.2742 | −0.1710 | 0.5724 | 0.0475 |
| Gumbel | $1.98 \times 10^{-6}$ | $1.27 \times 10^{-6}$ | 2.553 | 0.2781 |
| Clayton | $1.02 \times 10^{-6}$ | $6.33 \times 10^{-6}$ | 2.553 | 0.2781 |
| Frank | −0.2881 | −0.1952 | 0.3172 | 0.0411 |

　　It can be seen from Table 2 that the Spearman correlation coefficient and Kendall correlation coefficient of the Gumbel function and Clayton function were positive, which

was not consistent with the complementary characteristics of negative correlation between wind and photovoltaic energy. Compared with the t-distribution function, the normal distribution function, and the Frank function, the Spearman correlation coefficient and the Kendall correlation coefficient of the Frank function were the closest to the sample data. On the other hand, the Euclidean distance and the maximum distance were the smallest. This shows that the fitting effect of the Frank function was the best, and the established wind–PV complementary model fit more closely. Therefore, from the perspective of the fitting effect and the degree of fit, the Frank function fitting the wind–PV complementary relationship was the most representative.

Based on the wind power photovoltaic output historical data of the park in 2019, combined with the scenario generation method proposed in Equations (1)–(5) and considering that the whole year was divided into winter, summer, and a transition season, three typical daily wind and solar output scenarios were generated by the fuzzy clustering method. Among them, scenario 1 is summer, scenario 2 is winter, and scenario 3 is a transition season, as shown in Figures 6 and 7.

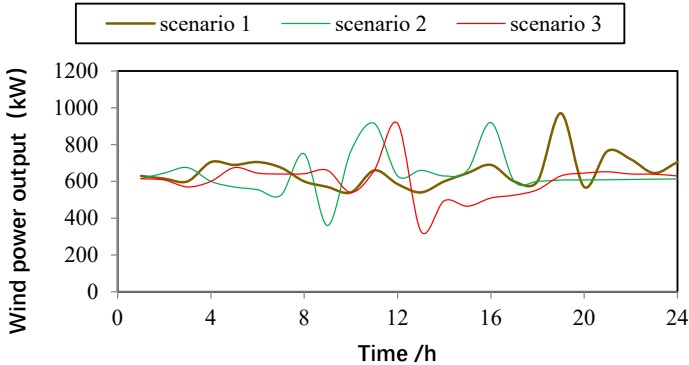

**Figure 6.** Typical wind power output scenarios.

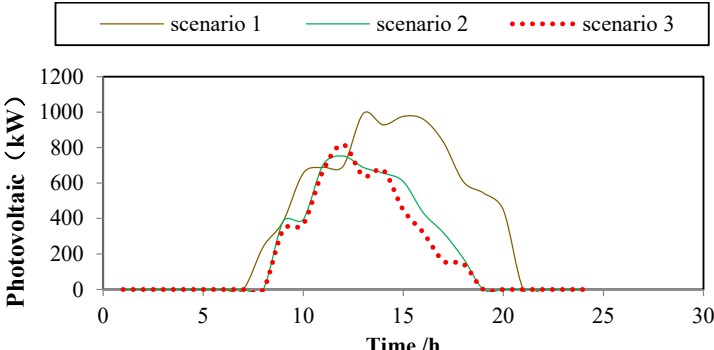

**Figure 7.** Typical photovoltaic output scenarios.

According to Figures 6 and 7, the trend of wind power and photovoltaic power generation was consistent or opposite in a certain period of time, with a certain correlation and complementarity. Meanwhile, wind power and photovoltaic output had obvious seasonal characteristics. The three typical daily output scenarios could effectively simulate the randomness and complementarity of the local wind and solar energy and improved the economic efficiency of the park scheduling.

(2) Wind power generation/photovoltaic power generation/total energy demand correlation result analysis.

Based on the load data of the park in 2019 combined with the stochastic fuzzy model, various loads in the park under different scenarios could be obtained, as shown in Figures 8–10.

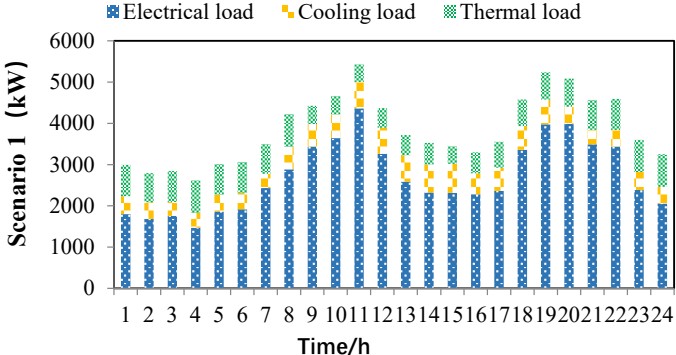

**Figure 8.** Energy demand in scenario 1.

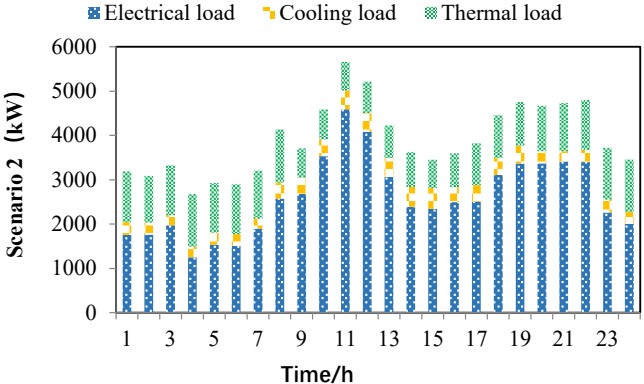

**Figure 9.** Energy demand in scenario 2.

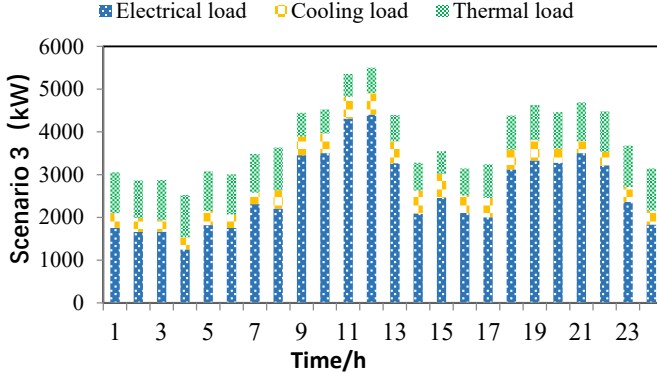

**Figure 10.** Energy demand in scenario 3.

According to Figures 8–10, the cooling energy demand of scenario 1 was relatively higher than that of scenarios 2 and 3. Since scenario 1 was in summer, there was more demand for cooling load. Compared with scenario 1 and scenario 3, the heating energy demand of scenario 2 was higher. As scenario 2 was winter, the heating demand of users increased. Scenario 3 was a transitional season with low heating and cooling load re-

quirements. Combining wind power, photovoltaic power generation, and total energy demand in different scenarios, the relationship between total output and total energy demand could be obtained as shown in Figure 11.

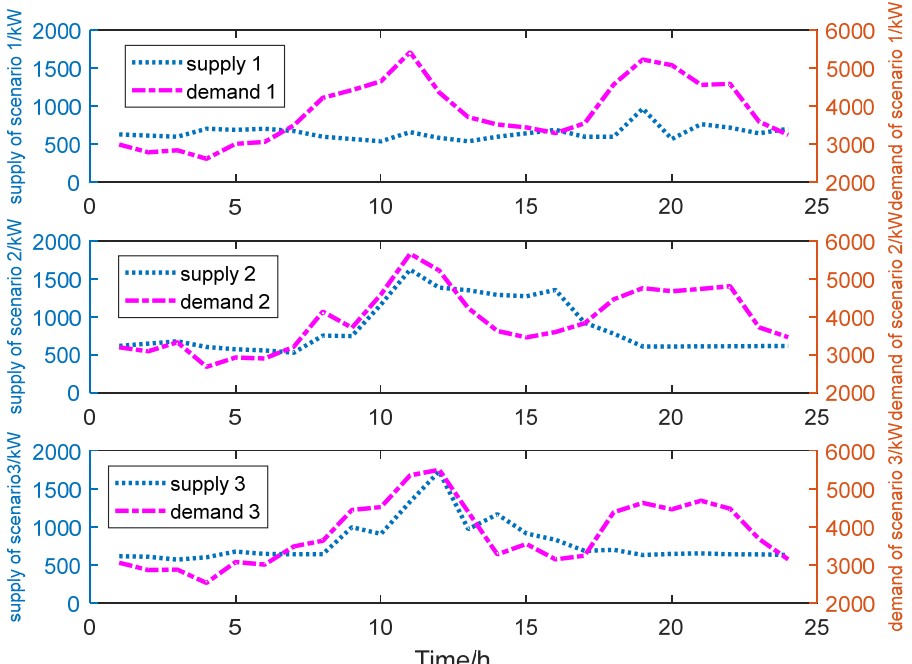

**Figure 11.** Supply and demand for different scenarios.

It can be seen from Figure 11 that the total energy demand presents a double peak; the first peak occurred between 11:00 and 13:00 and the total output value was also higher at this time. Wind power, photovoltaic power generation, and total energy demand were complementary. The second peak occurred at 18:00–21:00, when the output was small, and the complementarity between wind power generation, photovoltaic power generation, and total energy demand was weak. This is because from 11:00 to 13:00 the light intensity was high and the photovoltaic output was large, while from 18:00 to 21:00 the light intensity was almost zero, resulting in zero photovoltaic output.

(3) Analysis of the correlation results between electricity demand and electricity price.

Based on the electricity demand and electricity price data, the correlation results between the two are shown in Figure 12.

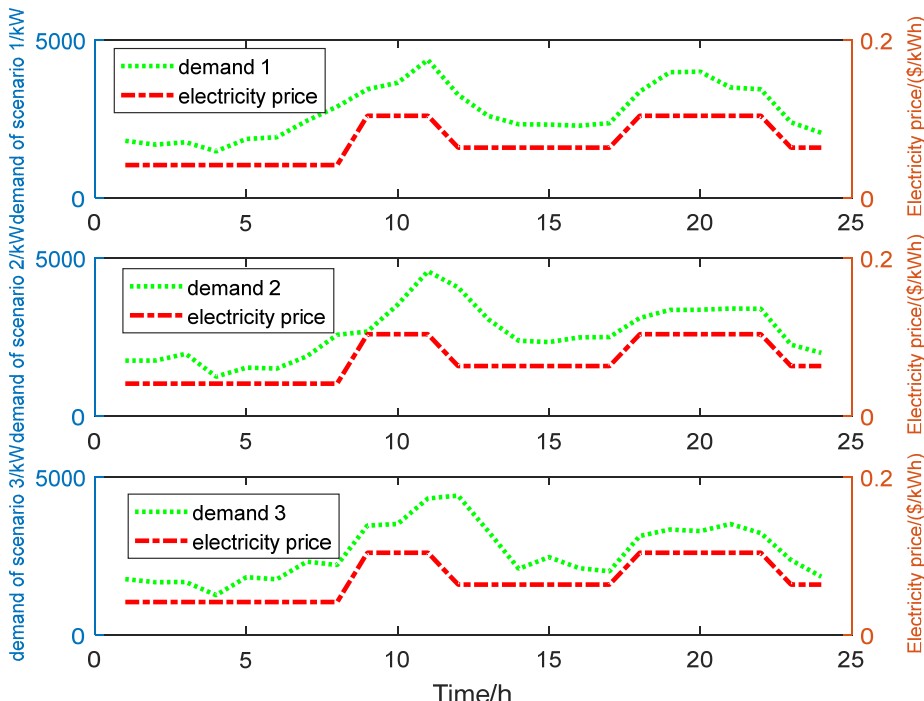

**Figure 12.** The correlation results between the electricity demand and electricity price.

It can be seen from Figure 12 that the peak periods of electricity demand were 9:00–11:00 and 18:00–22:00, the flat period was 12:00–17:00, and the valley period was 1:00–8:00. The time periods with a higher electricity price were 9:00–11:00 and 18:00–22:00, and the time period with a lower electricity price was 1:00–8:00. On the one hand, this shows that the random fuzzy model could effectively fit all kinds of energy demand and achieved a better match between all kinds of energy demand and the energy selling price. On the other hand, this shows that there was a strong correlation between electricity demand and electricity price.

(2) Scheduling optimization results of different scenarios.

Based on the dispatching optimization model, the wind power and photovoltaic output scenario and various energy demands of this paper are shown in Table 3.

**Table 3.** Park scheduling optimization results.

| Scenario | Load | Energy Production | | Energy Conversion | | | | | Energy Storage | | | Objective Function | |
|---|---|---|---|---|---|---|---|---|---|---|---|---|---|
| | | Gas Turbine (kWh) | Gas Boiler (kWh) | P2H (kWh) | P2C (kWh) | P2G (kWh) | H2C (kWh) | Battery (kWh) | Thermal Energy Storage (kWh) | Ice Storage Tank (kWh) | Gas Tank (kWh) | Profit ($) | Purchased Electricity (%) |
| Scenario 1 | Electricity | 51516 | _ | −2835 | 0 | −1215 | _ | −3854 | _ | _ | _ | 1261 | 0.97 |
| | Heat | 10895 | 8600 | 2835 | _ | _ | −8746 | _ | −1830 | _ | _ | 608 | _ |
| | Cooling | _ | _ | _ | _ | _ | 8746 | _ | _ | 3153 | _ | 167 | _ |
| Scenario 2 | Electricity | 50289 | _ | −2767 | 0 | −1186 | _ | −3762 | _ | _ | _ | 1231 | 0.97 |
| | Heat | 16343 | 12900 | 2767 | _ | _ | −8538 | _ | 2745 | _ | _ | 883 | _ |
| | Cooling | _ | _ | _ | _ | _ | 8538 | _ | _ | 3077 | _ | 163 | _ |
| Scenario 3 | Electricity | 51713 | _ | −2751 | 0 | −1179 | _ | −3699 | _ | _ | _ | 1274 | 0.95 |
| | Heat | 13619 | 10750 | 2751 | _ | _ | −6997 | _ | 2287 | _ | _ | 720 | _ |
| | Cooling | _ | _ | _ | _ | _ | 6997 | _ | _ | 2552 | _ | 134 | _ |

It can be seen from Table 3 that the electric energy of the park was mainly supplied by wind turbines, photovoltaic units, and gas turbines, and the battery was used for peak shaving. The thermal energy of the park was mainly supplied by gas turbines, gas boilers, and P2H, and HS was used for peak shaving. The cooling energy of the park was mainly supplied by H2C, and the ice storage tank was used for peak shaving. In the optimal operation of the park, P2C was almost zero. Since the cost of electric cooling was higher than that of hot to cooling, the park will give priority to the conversion of heat energy to cooling energy in order to improve economic benefits. In each scenario, the purchased electricity rate of the park was less than 1%, which indicates that the park could greatly promote the consumption of clean energy. The heat gain of scenario 2 was $883, which was higher than that of scenario 1 and scenario 3. The cooling energy income of scenario 1 was $167, which was higher than that of scenario 2 and scenario 3. The reasons for both of these cases were the seasons involved. Scenario 1 required more cooling energy in summer and scenario 2 required more heat energy in winter.

The results of the equipment scheduling optimization are shown in Figures 13–15.

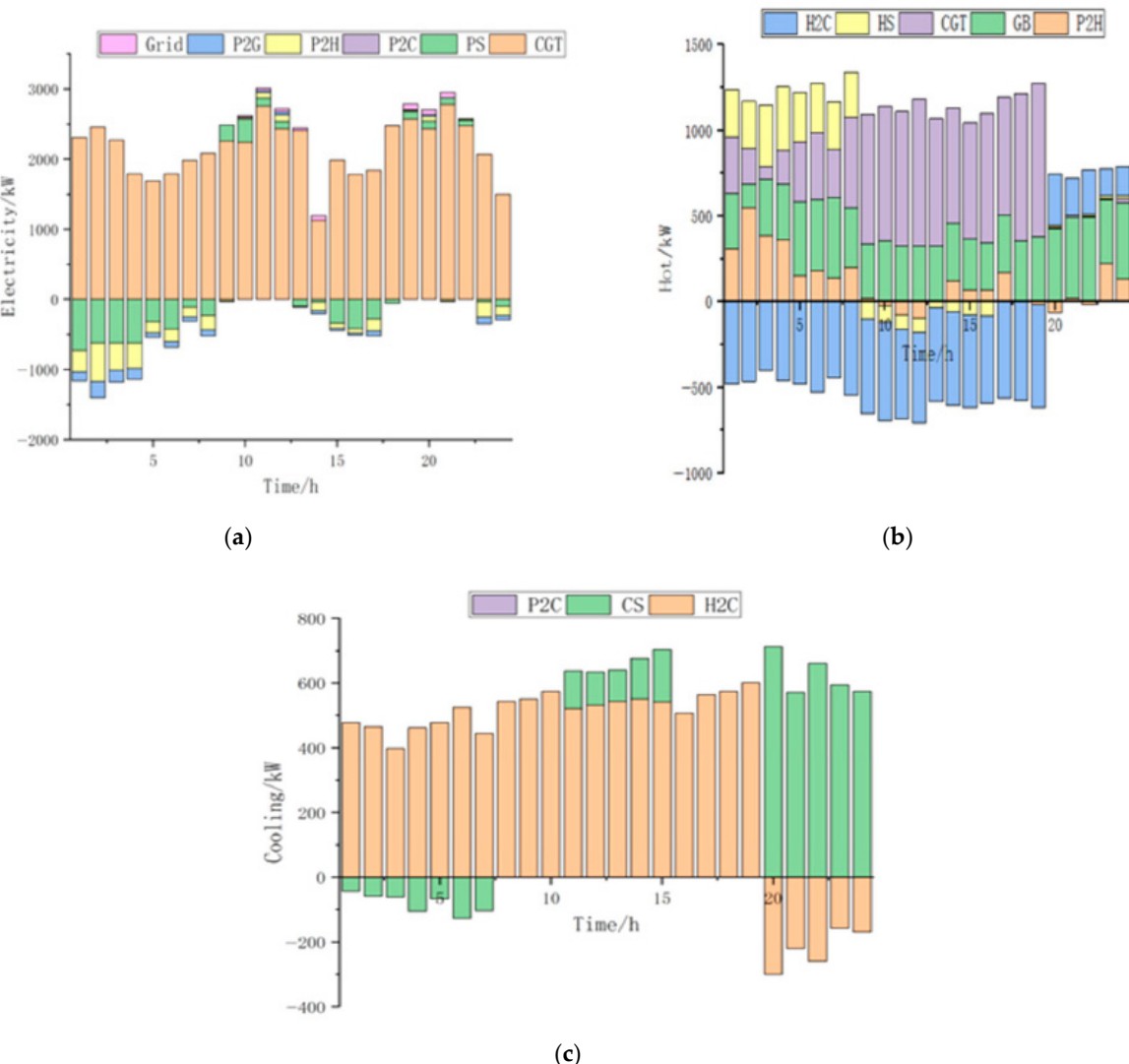

(a)

(b)

(c)

**Figure 13.** (**a**): Scenario 1 scheduling electricity load optimization results. (**b**): Scenario 1 scheduling hot load optimization results and (**c**) scenario 1 scheduling cooling load optimization results.

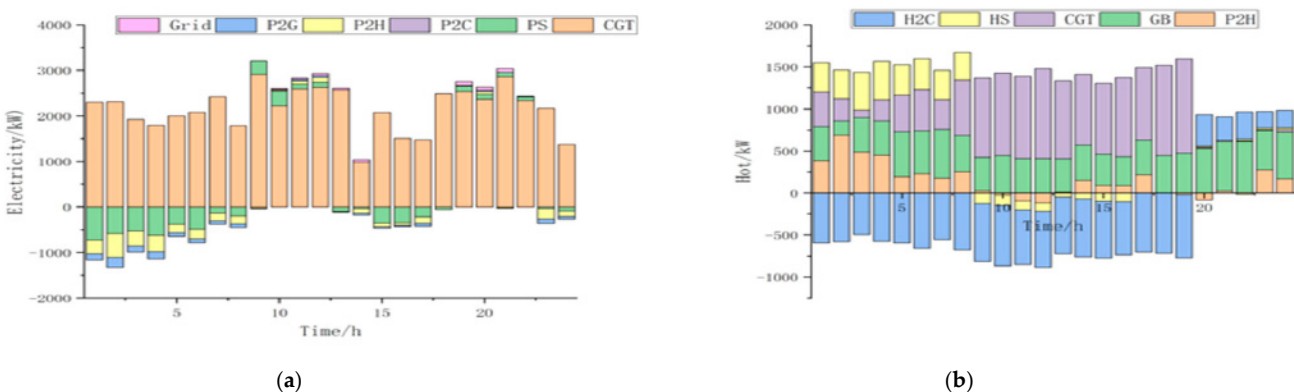

**Figure 14.** (**a**) Scenario 2 scheduling electricity load optimization results; (**b**) scenario 2 scheduling hot load optimization results; and (**c**) scenario 2 scheduling cooling load optimization results.

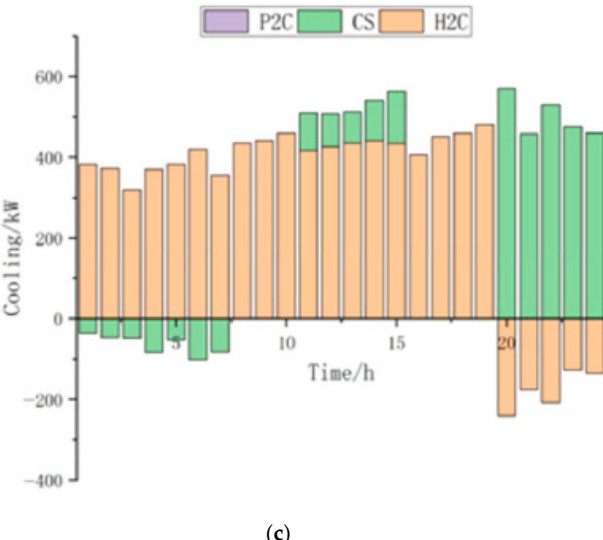

(**c**)

**Figure 15.** (**a**) Scenario 3 scheduling electricity load optimization results; (**b**) scenario 3 scheduling hot load optimization results; and (**c**) scenario 3 scheduling electricity cooling optimization results.

According to the figure, it can be seen from the power dispatching that in the period of low power demand the dispatching optimization of the production link converted the power into heat energy and natural gas through P2H and p2g. The scheduling optimization of the storage link stored the power through the storage battery for low storage and high release. It can be seen from the thermal energy scheduling that in the peak period of thermal energy the scheduling optimization of the production link converted the heat energy into cooling energy through H2C. The production link used thermal energy storage for low storage and high release. From the cooling energy scheduling, it can be seen that the cooling energy scheduling optimization used an ice storage tank for low storage and high release. The peak price of energy was higher than the low price, so the conversion link and storage link could cause an energy price difference through the time sequence of the transfer of energy.

(3) Comparative analysis of different scenario optimizations.

In order to deeply analyze the role of storage devices in the storage link and the different scheduling optimization behaviors between multi-objective and single-objective microgrids, four different scenarios were set up for a comparative analysis on a typical summer day—i.e., scenario 1. The scenario settings are shown in Table 4.

**Table 4.** Different settings.

|  | Scenario1 | Scenario 2 | Scenario 3 | Scenario 4 |
|---|---|---|---|---|
| Multiple objective | N | Y | N | Y |
| Energy storage device | N | N | Y | Y |

Note: Y means that the condition is met and N means that it is not. The energy storage device includes energy storage batteries, thermal energy storages, ice storage tanks, and air storage tanks.

The economic results of the park under different scenarios are shown in Table 5 and the environmental protection results are shown in Table 6, in which the change rates of the park income and the purchased electricity rate were compared with those of scenario 4.

**Table 5.** The economic results of the park under different scenarios.

| Scenario | Profit ($) | Change Rate of Profit (%) | Cost of Investment ($) | Cost of Operating ($) | Net Present Value ($) | Payback Period (years) |
|---|---|---|---|---|---|---|
| Scenario 1 | 1413.09 | −30.59 | 10,832.33 | 9042.21 | 11,019.88 | 11.03 |
| Scenario 2 | 1617.06 | −20.58 | 10,832.56 | 7929.07 | 12,824.23 | 10.88 |
| Scenario 3 | 1926.89 | −53.58 | 12,750.24 | 5721.14 | 14,758.91 | 10.25 |
| Scenario 4 | 2035.97 | − | 12,750.35 | 4083.66 | 15,784.07 | 9.64 |

**Table 6.** The environmental protection results of the park under different scenarios.

| Scenario | Purchased Electricity Rate (%) | $CO_2$ Emissions (t) | Change Rate of Purchased Electricity Rate (%) | $CO_2$ Emissions Change Rate (%) |
|---|---|---|---|---|
| Scenario 1 | 2.48 | 1.78 | −30.59 | 95.60% |
| Scenario 2 | 1.59 | 1.23 | −20.58 | 35.16% |
| Scenario 3 | 1.22 | 1.56 | −53.58 | 71.43% |
| Scenario 4 | 0.97 | 0.91 | − | — |

According to Table 5, the impact of energy storage devices on the park could be compared between scenario 1 and scenario 3 or scenario 2 and scenario 4. Among them, scenarios 1 and 2 had no energy storage devices and scenario 3 and scenario 4 were equipped with energy storage devices. Compared with scenario 3 and 4, from an economic point of view, although the investment costs of scenarios 1 and 2 were low, the operating costs and investment payback period were high, resulting in the low park revenue and net present value of scenario 1 and scenario 2. From the perspective of environmental protection, scenarios 1 and 2 had high purchased electricity and high $CO_2$ emissions values. This indicates that energy storage devices could improve the economic efficiency of the park and promote the consumption of renewable energy. On the one hand, this is due to the low storage and high release of energy storage devices, which cause the energy price difference, thus improving the economic efficiency of the park. On the other hand, the flexibility of energy storage and release of energy storage devices had a positive effect on maintaining the balance of supply and demand, reducing amount of electricity purchased in the park and promoting the consumption of renewable energy.

The impact of the multi-objective optimization on the park could be compared between scenario 1 and scenario 2 or scenario 3 and scenario 4, in which scenario 1 and 3 were single-objective and scenarios 2 and 4 were multi-objective optimizations. Compared with scenario 2 and 4, in scenario 1 and scenario 3, from the economic point of view, the operating costs were high, the investment recovery period was long, and the income and net present value were low. From the perspective of environmental protection, the purchased electricity and $CO_2$ emissions were greatly reduced. This indicates that it was more effective to promote the optimal scheduling of the park system and improve the economy and environmental protection of the park by considering both the park revenue and the purchased electricity rate.

**5. Conclusions**

Based on the uncertainty of wind power and photovoltaic output and load, a scenario generation model considering the correlation and uncertainty of wind and solar power and a stochastic fuzzy model of load uncertainty were constructed in this paper, and a multienergy complementary park scheduling strategy considering both economy and environmental protection was proposed. The proposed strategy had the following advantages:

(1) In this paper, based on the uncertainty of wind power and photovoltaic energy, the Copula function was used to generate a wind–solar complementary joint-distribution

function. Compared with the traditional Weibull distribution and Beta distribution, the typical daily scenario generated could more effectively fit the wind power and photovoltaic output characteristics of the park.

(2) Based on the uncertainty and fuzziness of the cooling and heating load affected by the subjectivity of users, a random fuzzy model was proposed to fit the cooling and heating load in the park using fuzzy numbers. The fitting results could effectively match the energy selling price in the peak, flat, and valley periods.

(3) On the one hand, considering the economy and environmental protection of the park, a scheduling optimization model of the multienergy complementary park was constructed. On the other hand, the allocation of various energy storage devices in the park could not only improve the park's income, but also promoted the consumption of renewable energy.

**Author Contributions:** Methodology, C.T.; Supervision, D.N. and X.G.; Writing—original draft, S.G.; Writing—review & editing, G.W. All authors have read and agreed to the published version of the manuscript.

**Funding:** The 2018 Key Projects of Philosophy and Social Sciences Research, Ministry of Education, China: 18JZD032 111 Project, Ministry of Science and Technology of People's Republic of China: B18021

**Institutional Review Board Statement:** Not applicable. for studies not involving humans or animals.

**Informed Consent Statement:** Not applicable.

**Data Availability Statement:** Publicly available datasets were analyzed in this study. This data can be found here: https://kns.cnki.net/kcms/detail/detail.aspx?FileName=SJES147D12446181E823FBB04B9629F034C4 &DbName=SJES2020; https://kns.cnki.net/kcms/detail/detail.aspx?FileName=DLXT201815019&DbName=CJFQ2018.

**Acknowledgments:** This work is supported by the 2018 Key Projects of Philosophy and Social Sciences Research, Ministry of Education, China (Project No. 18JZD032), 111 Project, Ministry of Science and Technology of People's Republic of China (Project No. B18021).

**Conflicts of Interest:** The authors declare no conflict of interest.

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
