# Peer review of "Multi-Objective Optimization of a Microgrid Considering the Uncertainty of Supply and Demand"

_sustainability, doi:10.3390/su13031320_

Round 1
Reviewer 1 Report
The work that has been performed and the manuscript preparation are considerably satisfactory.
The author clearly describes the uncertainty occurring in an hybrid source of solar and wind.
The manuscript describes the analytical method for estimating uncertainty in distribution loads, with applications to network monitoring. This helps the researcher to focus more on integrating the various energy resources.
The risk and the average operating reserve are evaluated which creates interest towards the researcher towards generation and distribution.
The multi-objective optimization is clearly described by various scenarios.
Improve the figure quality with proper axis representation.
Author Response
Dear editors and reviewers:
Thank you for your letter and for the reviewers’ comments concerning our manuscript entitled “Multi-objective optimization of microgrid considering the un-certainty of supply and demand (1068149)”. Those comments are all valuable and very helpful for revising and improving our paper, as well as the important guiding significance to our researches. We have studied comments carefully and have made correction which we hope meet with approval. Revised portion are marked in yellow in the paper. The main corrections in the paper and the responds to the reviewers’ comments are as following:
Point 1: Improve the figure quality with proper axis representation.
Response 1: Thanks for the valuable suggestions of experts. The expert’s advice is very constructive. According to the expert’s advice, we have changed the X-axis label of Figure 6 and Figure 7 from "time/t" to "time/h", and marked yellow in the text. Thanks again to the expert for your valuable suggestions.
Special thanks to you for your good comments!
Kind regards,
Shiping Geng and all authors

Reviewer 2 Report
This paper deals with uncertainties with renewable power generation and loads in the energy management system of small local grids. There are some comments.
In terms of terminology, "microgrid" is more common than "multi-energy complementary park system", then it is suggested to use the microgrid term or clarify the difference in the paper.
This issue has been extensively studied and the author should improve the literature review part in the introduction to highlight the existing gap and the motivation of using the proposed method, i.e. Copula function and fuzzy logic. it is strongly suggested to categorize the existing work into some groups and discuss their disadvantages and shortcomings.
Also, there are some works that considered the uncertainties using fuzzy logic and hybrid optimization that could be to added the literature list such as (but not limited to):
"A hybrid method for simultaneous optimization of DG capacity and operational strategy in microgrids considering uncertainty in electricity price forecasting." Renewable energy 68 (2014): 697-714.
"Operational strategy optimization in an optimal sized smart microgrid." IEEE Transactions on Smart Grid 6, no. 3 (2014): 1087-1095.
Please correct the X-axis label in the figures in the result section it should be: Time (hours)
Author Response
Dear editors and reviewers:
Thank you for your letter and for the reviewers’ comments concerning our manuscript entitled “Multi-objective optimization of microgrid considering the un-certainty of supply and demand (1068149)”. Those comments are all valuable and very helpful for revising and improving our paper, as well as the important guiding significance to our researches. We have studied comments carefully and have made correction which we hope meet with approval. Revised portion are marked in yellow in the paper. The main corrections in the paper and the responds to the reviewers’ comments are as following:
Point 1: In terms of terminology, "microgrid" is more common than "multi-energy complementary park system", then it is suggested to use the microgrid term or clarify the difference in the paper.
Response 1: Thanks for the valuable suggestions of expert. The expert’s advice is very correct. According to experts’ suggestions, we checked the full text, replaced "multi-energy complementary park system" with "microgrid", and marked the article as yellow
Point 2: This issue has been extensively studied and the author should improve the literature review part in the introduction to highlight the existing gap and the motivation of using the proposed method, i.e. Copula function and fuzzy logic. it is strongly suggested to categorize the existing work into some groups and discuss their disadvantages and shortcomings. Also, there are some works that considered the uncertainties using fuzzy logic and hybrid optimization that could be to added the literature list such as (but not limited to):"A hybrid method for simultaneous optimization of DG capacity and operational strategy in microgrids considering uncertainty in electricity price forecasting." Renewable energy 68 (2014): 697-714. "Operational strategy optimization in an optimal sized smart microgrid." IEEE Transactions on Smart Grid 6, no. 3 (2014): 1087-1095.
Response 2: Thanks for the valuable suggestions of expert. Expert advice is very constructive. According to the suggestion of expert , in the introduction, the existing research on the uncertainty and fuzziness of the two ends of the source and load and the advantages of the copula function and fuzzy logic model proposed in this paper have been supplemented and marked yellow in the text, which are mainly divided into the following two aspects:
(1) For the uncertainty of wind and photovoltaic output, most literatures use robust optimization method, but this method can not expand the random correlation between two or more variables, and the Copula solution model proposed in this paper effectively solves this problem.
(2) For the stochastic fuzzy model proposed in this paper, it is considered that in the traditional energy uncertainty processing methods, stochastic programming model, cloud model theory, interval estimation, but they ignore the economic impact in the process of processing, and the stochastic fuzzy model can effectively avoid this problem and reduce the deviation between the predicted value and the actual value. The relevant documents are as follows:
[1] Mohammad H, Moradi,Mohsen E. A hybrid method for simultaneous optimization of DG capacity and operational strategy in microgrids considering uncertainty in electricity price forecasting[J]. Renewable Energy,2014,68.
[2] Moradi Mohammad H , Eskandari M , Mahdi Hosseinian S. Operational strategy optimization in an optimal sized smart microgrid. IEEE Transactions on Smart Grid 6, no. 3 (2014): 1087-1095.
[3] Wang YW, Tang L, Yang YJ, et al. A stochastic-robust coordinated optimization model for CCHP micro-grid considering multi-energy operation and power trading with electricity markets under uncertainties[J]. Energy,2020,198.
[4] Amir Aris Lekvan,Reza Habibifar,Mehran Moradi,Mohammad Khoshjahan,Sayyad Nojavan,Kittisak Jermsittiparsert. Robust Optimization of Renewable-based Multi-Energy Micro-Grid Integrated with Flexible Energy Conversion and Storage Devices[J]. Sustainable Cities and Society,2020.
[5] Lojowska Alicja, Kurowicka Dorota, Papaefthymiou Georgios, Sluis Lou. Stochastic modeling of power demand due to EVs using copula. IEEE Trans Power Syst 2012;27(4):1960–7.
[6] Pashajavid E, Golkar MA. Non-Gaussian multivariate modeling of plug-in electric vehicles load demand. Int J Electr Power Energy Syst 2014;61:197–207.
[7] Valizadeh Haghi H, Tavakoli Bina M, Golker MA, Moghaddas-Tafreshi SM. Using copulas for analysis of large datasets in renewable distribution generation: PV and wind power integration in Iran. Renew Energy 2010;35(9):1991–2000.
[8] Lin L, Zhang Y, Gu J. Research on Uncertainty of Incentive Regional Flexible Load Response Based on Cloud Mod-el[J].Power System Technology,2020,44(11):4192-4201.
[9] Yi WF, Zhang YW, Zeng B,et al.Multi-form incentive demand-side response coordination to balance the robust optimization configuration of renewable energy fluctuations[J]. Transactions of China Electrotechnical Society, 2018,33(23):5541-5554.
[10] Cui Y, Zhou HJ, Zhong WZ, et al. Considering the uncertainties on both sides of source and load, low-carbon dispatch of wind power power system[J].Electric Power Automation Equipment,2020,40(11):85-93.
[11] Ji L, Niu DX. Xu M, et al. An optimization model for regional micro-grid system management based on hybrid inexact stochastic-fuzzy chance-constrained programming[J]. International Journal of Electrical Power and Energy Sys-tems,2015,64.
Point 3: Please correct the X-axis label in the figures in the result section it should be: Time (hours)
Response 3: Thanks for the valuable suggestions of experts. The expert’s advice is very constructive. According to the expert’s advice, we have changed the X-axis label of Figure 6 and Figure 7 from "time/t" to "time/h", and marked yellow in the text. Thanks again to the expert for your valuable suggestions.
Special thanks to you for your good comments!
Kind regards,
Shiping Geng and all authors

Reviewer 3 Report
The article subject is interesting and relevant but their are some issues that should be addressed:
- Abstract is confused, the authors should better explain their terminology and perform a comprehensive link between their text. You should explain what do you mean with "park" in the abstract, readers can only understand it in the Introduction;
- The authors should also avoid the use of double synonyms like "load demand", adapt one;
- With "source load", do you mean supply and load/demand, or just load?
- Along the text the authors use different terminologies to refer to the same things, you should keep a coherency along the text. While source and load is used more technically, I suggest you to use supply and demand to achieve a broader audience;
- Avoid the use of very long phrases without connecting them as "virtual power plant(VPP) operation /considering a/ flexible risk avoidance model";
- There is a lack of definite and indefinite articles along the text to increase its fluidity and comprehension;
- In Fig. 1 when you refer "Supply", I would suggest to change to "Resource" or other synonim, to avoid confusion with "energy supply" previously used in other context;
- "Minimize the amount of purchased electricity to promote the consumption of clean energy" cannot be evaluated by this system, since the authors use gas systems and are not extrapolating the share of clean energy they purchase. So the goal seems to promote sustainability, avoiding the need to purchase electricity;
- Figures 4(a) and 4(b) are barely readable, I suggest to use monotone curves;
- In Figure 5 the authors should clarify why the gas price is so volatile in a daily basis contrary to reality;
- Considering the goal of maximizing profit and minimizing the purchase of electricity, the authors should study the wind-PV-total load/electricity price-park load complementarily and not just the wind-PV complementarily;
- Unfortunatelly, considering the three indicated contributions of the paper, the paper has lack of evidence in relation to the third objective: "economy and environmental protection". The authors prove the energetic sustainability of the system as can be seen in Table 5, by considering at worse the park only purchase 2.48% of electricity. But, environmentally, the authors should indicate at least the CO2 emissions reduction by using this system. Economically, the authors should indicate the investment and operational costs of the four scenarios presented in Table 5, and at least, compute the cash-flows, the net present value and the payback period in relation to purchasing electricity.
Author Response
Dear editors and reviewers:
Thank you for your letter and for the reviewers’ comments concerning our manuscript entitled “Multi-objective optimization of microgrid considering the un-certainty of supply and demand (1068149)”. Those comments are all valuable and very helpful for revising and improving our paper, as well as the important guiding significance to our researches. We have studied comments carefully and have made correction which we hope meet with approval. We have carefully revised and improved the language of the full text.Revised portion are marked in yellow in the paper. The main corrections in the paper and the responds to the reviewers’ comments are as following:
Point 1: Abstract is confused, the authors should better explain their terminology and perform a comprehensive link between their text. You should explain what do you mean with "park" in the abstract, readers can only understand it in the Introduction;
Response 1: Thanks for the valuable suggestions of experts. Based on the valuable suggestions of experts, we reorganized the abstract and explained the meaning of the park. The park refers to a microgrid that gathers distributed energy such as wind and photovoltaics and has requirements for cold, heat and electricity at the same time. Related changes have been marked in yellow in the article.
Point 2: The authors should also avoid the use of double synonyms like "load demand", adapt one
Response 2: Thanks for the valuable suggestions of experts. According to the reviewer’s comments. In order to avoid the use of double synonyms, we uniformly revised the "load demand" in the full text to "energy demand".
Point 3: With "source load", do you mean supply and load/demand, or just load?
Response 3: Thanks for the valuable suggestions of experts. For the term "source load", it includes both supply side and demand side.
Point 4: Along the text the authors use different terminologies to refer to the same things, you should keep a coherency along the text. While source and load is used more technically, I suggest you to use supply and demand to achieve a broader audience;
Response 4: Thanks for the valuable suggestions of experts. The expert’s advice is very targeted. According to the expert’s advice, in order to expand the audience, we use the relationship of supply and demand to indicate the power and the load in the article. Related changes have been marked in yellow in the article.
Point 5: Avoid the use of very long phrases without connecting them as "virtual power plant(VPP) operation /considering a/ flexible risk avoidance model";
Response 5: Thanks for the valuable suggestions of experts. Considering the reviewer’s suggestion , In order to avoid using very long phrases, we have checked the full text and marked the relevant changes in yellow. As follows:
(1) In the introduction,”virtual power plant(VPP) operation flexible risk avoidance model” has changed into “virtual power plant (VPP) considering an operation flexible risk avoidance model”.
Point 6: There is a lack of definite and indefinite articles along the text to increase its fluidity and comprehension;
Response 6: Thanks for the valuable suggestions of experts. According the suggestion of the experts. In order to enhance the fluidity and comprehension of the article, we read through the article, supplemented the lack of indefinite articles and articles, and marked the article as yellow
Point 7: In Fig. 1 when you refer "Supply", I would suggest to change to "Resource" or other synonim, to avoid confusion with "energy supply" previously used in other context;
Response 7: Thanks for the valuable suggestions of experts. The suggestions of experts is really true. According to the expert’s suggestion, we modified Fig.1 and changed the "Resource" in Fig.1 to "energy supply"
Point 8: "Minimize the amount of purchased electricity to promote the consumption of clean energy" cannot be evaluated by this system, since the authors use gas systems and are not extrapolating the share of clean energy they purchase. So the goal seems to promote sustainability, avoiding the need to purchase electricity;
Response 8: Thanks for the valuable suggestions of experts. The suggestions of experts is really true. This article assumes that the purchased electricity is supplied by conventional generators such as thermal power generators. Therefore, under the condition of certain energy demand, the less the purchased electricity, the more clean energy the system consumes. Therefore, minimizing purchased electricity can promote sustainable development on the one hand, and promote clean energy consumption on the other.
Point 9: Figures 4(a) and 4(b) are barely readable, I suggest to use monotone curves
Response 9: Thanks for the valuable suggestions of experts. In order to increase the readability of Figure 4 (a) and Figure 4 (b), we not only modified Figure 4 (a) and Figure 4 (b) to bar graphs, but also added information about Figure 4 (a) and Figure 4 (b). The description of Figure 4 (a) and Figure 4 (b) is as follows:
- Among them, the X-axis label of Figure 4a is the number of hours in a year, a total of 8760h, and the Y-axis label is the wind speed at different times.
The X-axis label of Figure 4b is also the number of hours in a year, a total of 8760h, and the Y-axis is the light intensity at different times.
Point 10: In Figure 5 the authors should clarify why the gas price is so volatile in a daily basis contrary to reality
Response 10: Thanks for the valuable suggestions of experts. The expert’s advice is very correct. As experts said, in reality the price of natural gas is indeed relatively fixed, but in order to fully reflect the dispatching operation and energy flow in the system, similar to the price of electricity, heat, and cold, this article considers that natural gas also has a time-of-use price. Many scholars have carried out research on the application of natural gas time-of-use tariffs in integrated energy systems. Zhang. et al [1] considered the integrated demand response in the integrated energy system and constructed a gas-electric joint time-of-use price. Zhang. et al[2] divided natural gas into three periods of peak, flat and valley, and applied the time-of-use electricity price of natural gas to the optimal scheduling of virtual power plants connected to gas and electricity. Pan. et al [3] considered the time-of-use price of natural gas to be used in the economic dispatch of a virtual power plant containing wind-light-gas-storage. Thus the gas price is so volatile in a daily. The relevant refrence are as follows:
[1] Zhang X.H., Huang W., Liu Q., Wang F. Gas-electricity joint time-of-use pricing optimization model based on comprehensive demand response [J]. Proceedings of the CSU-EPSA, 2019, 31(04): 91-98.
[2] Zhang J.L. Fan W., Tan Z.F., De G.J.R.F., Yang S.B., Sun J.X. Multi-objective scheduling optimization model of gas-electric interconnection virtual power plant in consideration of demand response [J]. Electric Power Construction , 2020, 41(02): 1-10.
[3] Pan H., Liang Z.F., Xue Q.Z., Zheng F., Xiao Y.H. Economic dispatch of virtual power plant containing wind-light-gas-storage based on time-of-use electricity price [J]. Acta Energiae Solaris Sinica, 2020, 41(08): 115-122.
Point 11: Considering the goal of maximizing profit and minimizing the purchase of electricity, the authors should study the wind-PV-total load/electricity price-park load complementarily and not just the wind-PV complementarily;
Response 11: Thanks for the valuable suggestions of experts. The suggestions of experts is really true. Expert advice is very constructive. On the basis of the original wind and PV complementarity, study of the wind-PV-total load/electricity price-park load complementarily has been added. The additional part has been marked yellow in the text. As follows:
(1) It can be seen from Figure 11 that the total energy demand presents a double peak; the first peak occurs between 11:00 and 13:00 and the total output value is also higher at this time. Wind power, photovoltaic power generation, and total energy demand are complementary. The second peak occurs at 18:00-21:00, when the output is small, and the complementarity between wind power generation, photovoltaic power generation, and total energy demand is weak. This is because from 11:00 to 13:00 the light intensity is high and the photovoltaic output is large, while from 18:00 to 21:00 the light intensity is almost zero, resulting in zero photovoltaic output.
(2) It can be seen from Figure 12 that the peak periods of electricity demand are 9:00-11:00 and 18:00-22:00, the flat period is 12:00-17:00, and the valley period is 1:00-8: 00. The time periods with a higher electricity price are 09: 11:00 and 18:00-22:00, and the time period with a lower electricity price is 1:00-8:00. On the one hand, this shows that the random fuzzy model can effectively fit all kinds of energy demand and achieve a better match between all kinds of energy demand and the energy selling price. On the other hand, this shows that there is a strong correlation between electricity demand and electricity price.
Point 12: Unfortunatelly, considering the three indicated contributions of the paper, the paper has lack of evidence in relation to the third objective: "economy and environmental protection". The authors prove the energetic sustainability of the system as can be seen in Table 5, by considering at worse the park only purchase 2.48% of electricity. But, environmentally, the authors should indicate at least the CO2 emissions reduction by using this system. Economically, the authors should indicate the investment and operational costs of the four scenarios presented in Table 5, and at least, compute the cash-flows, the net present value and the payback period in relation to purchasing electricity.
Response 12: Thanks for the valuable suggestions of experts. The suggestions of experts is very true. According to experts' suggestions, the scheduling optimization results of the four scenarios are divided into two aspects: economy and environmental protection. In terms of economy, indicators such as investment cost, operating cost, net present value and investment payback period have been added. In terms of environmental protection, two indicators of carbon dioxide emissions and the rate of change of carbon dioxide emissions have been added. Related changes have been marked yellow in the text.
Special thanks to you for your good comments!
Kind regards,
Shiping Geng and all authors

Round 2
Reviewer 3 Report
I would like to congratulate the authors for the clarity and technical improvements along the article.
Consider the following minor issues:
- Is hard to analyse Figures 4a and 4b, so, to evaluate the wind power and light intensity distributions I suggest to use decreasing monotonic curves (sorted from the highest to the lowest values);
- Completely check the english language and style, once after acceptance you only have the chance to fix minor errors.
Author Response
Dear editors and reviewers:
Thank you for your letter and for the reviewers’ comments concerning our manuscript entitled “Multi-objective optimization of a microgrid considering the uncertainty of supply and demand (1068149)”. Those comments are all valuable and very helpful for revising and improving our paper, as well as the important guiding significance to our researches. We have studied comments carefully and have made correction which we hope meet with approval. Revised portion are marked in yellow in the paper. The main corrections in the paper and the responds to the reviewers’ comments are as following:
Point 1: Is hard to analyse Figures 4a and 4b, so, to evaluate the wind power and light intensity distributions I suggest to use decreasing monotonic curves (sorted from the highest to the lowest values)
Response 1: Thanks for the valuable suggestions of experts. Based on the valuable suggestions of experts, We have revised Figure 4 a and b, and changed the bar graph to a curve to improve the readability and clarity of the graph, and marked in yellow in the article. Thanks again!
Point 2: Completely check the english language and style, once after acceptance you only have the chance to fix minor errors.
Response 2: Thanks for the valuable suggestions of experts. According to the reviewer’s comments, we read through the full text and modified some of the language expressions, and marked in yellow in the article. Thanks again!
Special thanks to you for your good comments!
Kind regards,
Shiping Geng and all authors
